# Stock Market Contagion during the Global Financial Crises: Evidence from the Chilean Stock Market

**Sakthi Mahenthiran** [1],*, **Tom Gjerde** [2] **and Berta Silva** [3]

[1]   Lacy School of Business, Butler University, Indianapolis, IN 46208, USA
[2]   Clark H. Byrum School of Business, Marian University, Indianapolis, IN 46222, USA; tgjerde@marian.edu
[3]   Escuela de Comercio, Pontificia Universidad Católica de Valparaíso, Valparaíso 2340031, Chile;
      berta.silva@pucv.cl
*   Correspondence: smahenth@butler.edu; Tel.: +1-317-940-8024

**Abstract:** The study examines evidence for the transmission of the US and EU financial crises via investor holdings into the Chilean stock market following two global financial crises, in 2008 and 2011. The study modified the models of Bekaert et al. (2014), and Dungey and Gajurel (2015) on the 2007–2009 global financial crisis and extends the period to include the European debt crisis of 2010–2011. The study produced three main contributions. First, changes in the equity holdings of retail investors were a key source of contagion following the 2008 US financial crisis. Second, investor herding during the 2011 financial crisis is shown to be low based on the co-movement of equity holdings between the four investor groups studied. Third, investor behavior during the 2011 EU crisis differs from that of the 2008 US financial crisis, which we attribute to firms in Chile adopting international financial reporting standards (IFRS) and improving their corporate governance. We compared the findings to the prior contagion studies that rely on Chilean return data to highlight the contributions to international financial research, particularly as it relates to the functioning of emerging capital markets during financial crises.

**Keywords:** stock market contagion; global financial crises; Chilean stock market; investor herding; investor holdings; emerging capital markets; corporate governance

**JEL Classification:** G11; G15; G40

## 1. Introduction

Investor herd behavior is defined as an intention of investors to mimic the behavior of other investors. This is a psychological behavior of market participants, and finance researchers have detected the presence of herding by looking at the relationship between individual firm stock returns and the average market returns. It is argued that investors ignore the fundamental analysis to explain stock prices' movements and instead base their decisions on aggregate market behavior, which has been found to be the case during the periods of large market movements (Chang et al. 2000; Bui et al. 2018). This may be the case with the coronavirus-related financial shock that is rattling the global financial markets in 2020. Hence, to study the 2020 global financial crises, it would be important to gain a better understanding of the reasons for the 2008 and 2011 crises. Bui et al. (2018) found that investor herd behavior is driven by both up and down overall market returns and showed that the variations in US stock market returns (not the Hong Kong stock market) were responsible for herd behavior in the frontier market of the Vietnamese stock exchange. Bekaert et al. (2014) and Dungey and Gajurel (2015) addressed the shortcoming in the literature on the co-movement of equity market returns surrounding the 2008 US mortgage crisis. The term "spillover effect" refers to the historical or expected cross-border

co-movement in asset prices, while the term "contagion" is reserved for unexpected or excessive spillover and is typically associated with negative shocks to the global financial system, for example, the 2020 global shock related to the coronavirus pandemic. The tendency of investors to sell in unison during a global financial market panic is termed investor "herding" and provides a potential explanation for a portion of the observed global co-movement in asset prices. Negative aspects of this type of contagion can be severe and include a reduction in the benefits from portfolio diversification, adverse effects on wealth and economic growth, and greater risk management issues for investors and policy makers (Connolly and Wang 2000; Kyle and Xiong 2001).

This study differs from previous contagion studies by examining changes in investors' equity holdings instead of equity returns. The objective is to determine if the rebalancing of equity portfolios by insiders, institutional investors, and retail investors is useful for studying the transmission of spillover, contagion, and investor herding during a global financial crisis (referred to as a GFC) into an emerging capital market. Although the methodology can be applied to any country, the current study examines the Chilean equity market during the period 2007–2013, a period that includes two GFC's—the US mortgage crisis and the EU debt crisis. Studies by Bekaert et al. (2014) and Dungey and Gajurel (2015) spanned the period 2007–2009 and did not include an analysis of the EU debt crisis. A synthesis of those two studies suggests that the Chilean equity market response to the US mortgage crisis was driven by excess movement in Chilean equity returns relative to expected returns based on company fundamentals. This type of contagion is referred to as "domestic" contagion by Bekaert et al. (2014) and as "idiosyncratic" contagion by Dungey and Gajurel (2015).

The additional feature of the 2007–2013 study period is that it includes the adoption of international financial reporting standards (IFRS) by Chilean firms in 2009, which required the voluntary adoption of fair value accounting standards during the period 2009–2013. Firms in the study began voluntarily adopting fair value standards in 2009, introducing the possibility that changes in the reporting environment may explain changes in equity holdings, given that the US mortgage crisis occurred pre-adoption and the EU debt crisis occurred post-adoption.

In emerging markets, insiders are long-term investors with superior information on the value of the firm relative to outside investors. Hence, if a GFC causes firm value to deviate from intrinsic value, then insiders might see it as an opportunity to increase their equity investment positions during a period in which outsiders are reducing their equity holdings. On the other hand, since insiders already control the majority of the equity shares, they may not feel the urge to interfere and "catch a falling knife" when their own company share prices are tumbling. Therefore, a critical tradeoff that arises during a GFC is the degree to which informed insiders are willing to ignore a temporary increase in risk and uncertainty and step-in to purchase shares they view as temporarily undervalued in an emerging market. A reason this tradeoff is crucial is that without insider buying, emerging markets with highly concentrated ownership may become illiquid during episodes of panic selling during a GFC. However, if informed insiders view these departures from intrinsic value as a buying opportunity, then they serve an invaluable role as liquidity providers to the market during periods of contagion. Additionally, this role of insiders would be more important in an emerging capital market than in a developed capital market because of the limited flexibility of the central bank and security regulators to avert the effects of a GFC.

The US is the top foreign source of investment in Chilean equity and investment fund shares, by value of holdings. Following the US, the EU countries represent seven of the next nine sources of foreign investment in Chilean equity and investment fund shares[1]. Therefore, international investors in Chilean securities introduce a mechanism through which foreign shocks can enter Chile's financial markets. Aguiar and Gopinath (2005) suggest that financial panics create illiquid markets when foreign

---

[1]　From Coordinated Portfolio Investment Survey 2013, IMF Table 13: Portfolio Investment Liabilities: Top Ten Economies by Size of Liabilities/ equity and investment fund shares, for Chile. http://data.imf.org/regular.aspx?key=32986.

sellers flood the market with sell orders at "fire sale" prices and flee less developed emerging markets for safe havens in developed countries. Hence, an adverse economic shock in the US or EU may force international retail investors and mutual fund managers to sell their Chilean equity holdings to meet margin calls and redemption requests. In this manner, changes in the Chilean equity holdings of international investors could play an important role in the transmission of a foreign-sourced financial crisis into the Chilean stock market.

A source of significant local institutional investment in Chile is its system of private pension funds. Chilean law limits the level of foreign securities that pension funds can hold. Thus, the exposure of local pension funds to foreign equity markets is limited, thereby reducing the ability of pension funds to serve as a transmission mechanism for foreign-sourced crises and contagions.

According to Gillian and Starks (2003), the typical retail investor is less sophisticated than institutional fund managers are, even though retail investors may be rational in the short-term during a GFC. Furthermore, retail investors include both domestic and foreign investors, and may be more prone to panic selling and herding behavior during a GFC relative to local institutional investors. Therefore, we explore the possibility that retail investors reduce their holdings of Chilean companies during a GFC, which exacerbate any downward pressure on the value of equity shares traded on the Chilean stock exchange, now referred to as the Santiago stock exchange.

Given the potential for insiders, institutional investors, and retail investors to magnify or mitigate spillovers and contagion, we explore the relationship between the two recent past GFC's, contagion effects, and creditor monitoring as a corporate governance mechanism on the equity holdings during the 2007–2013 study period. Our study departs from the existing literature on herding behavior and contagion in a number of ways. First, we examine the relationship between equity market holdings of investors instead of equity market returns. Our focus on equity holdings, in contrast to equity returns, introduces the possibility of identifying the specific groups of investors who serve as a mechanism for the transmission of foreign-sourced financial shocks into an emerging capital market. Second, we introduce EU equity returns as an additional source of spillover and contagion allowing us to study two crises, the US mortgage crisis and the EU debt crisis. Interestingly, the adoption of IFRS by Chilean companies occurred between the two GFC's and provides an opportunity to consider whether differing responses of the investor groups to the GFC's are attributable to IFRS adoption. Third, the use of investor holdings rather than return models introduces a novel method of studying contagion and herding among investor groups that can be generalized to studies of other countries and future global financial crises.

In the current study, we view retail investors as noise traders, pension funds and insiders as long-term value-based traders, and mutual funds as convergence traders, who may or may not be able to distinguish liquidity shocks from shocks to company fundamentals. Hence, we posit that in emerging capital markets, convergence traders exaggerate contagion effects if long-term investors, such as insiders or pension funds, fail to provide the same level of liquidity that they normally provide during a non-crisis period. Further, the link between equity market returns and aggregate household wealth creates a means for contagion to have negative and persistent effects on the real economy due to the behavior of retail investors. For example, Yuan (2005) found that wealth effects associated with investors can persist even if only a small fraction of investors experience borrowing constraints.

The study's findings support the idea that retail investors play a key role in the spread of contagion into the Chilean stock market. We measure changes in equity holdings across investor groups to show that herding behavior is not an important contributor to the lagged effects of the contagion. The result demonstrates the potential for our holdings-based methodology that can complement the traditional return-based methodology of Bekaert et al. (2014), Dungey and Gajurel (2015) and Bui et al. (2018). The study results highlight the importance of effective stock market regulation, banking regulation, and creditor monitoring for mitigating the effects of global financial crises.

The remainder of the study is organized as follows. The second section provides an outline of the Chilean institutional context exploring investor behaviors, the third section reviews the literature,

and the fourth section describes the methodology leading to the development of five hypotheses, and describes the sample, variable measurements, and the models tested. Section 5 presents the results, and Section 6 discusses investor behaviors and concludes the study by highlighting its limitations and future research directions.

## 2. Chilean Institutional Context

Chile's private pension system is characterized by six pension funds, referred to as "Adminstradoras de Fondos de Pensiones", or AFP. They have restrictions on their investments, including the size of their investment holdings in foreign securities[2]. They manage the pension accounts of the citizenry who invest in them through payroll deduction made by employers. According to Morales et al. (2013, p. 181), the AFP system had a positive effect in relation to, "(i) the emergence of reform in the legal system and the improvement of the oversight under which firms operate, which have influenced the quality of the external mechanisms regulating corporate governance, (ii) the emergence of greater capital market liquidity and trading volumes, and (iii) the professionalization of financial intermediaries". For these reasons, the Chilean stock market is considered a highly integrated Latin American market compared to most emerging country markets, such as the Vietnamese stock market studied by Bui et al. (2018). This fact makes the Chilean stock market an ideal emerging capital market to study the effects of GFC on investor holdings.

Figure 1 depicts Chilean, US, and EU annual GDP growth, and highlights the cross-country variation of GDP during the GFC. Figure 2 displays the change in selected components of Chilean GDP. It is noteworthy that the growth rate of Chilean total private consumption and GDP were impacted less by the GFC's relative to the substantial and negative growth in fixed capital formation and total domestic spending (i.e., GDP less imports). Hence, the impact of the GFC on Chile's economy were varied, yet substantial, on certain components of the GDP.

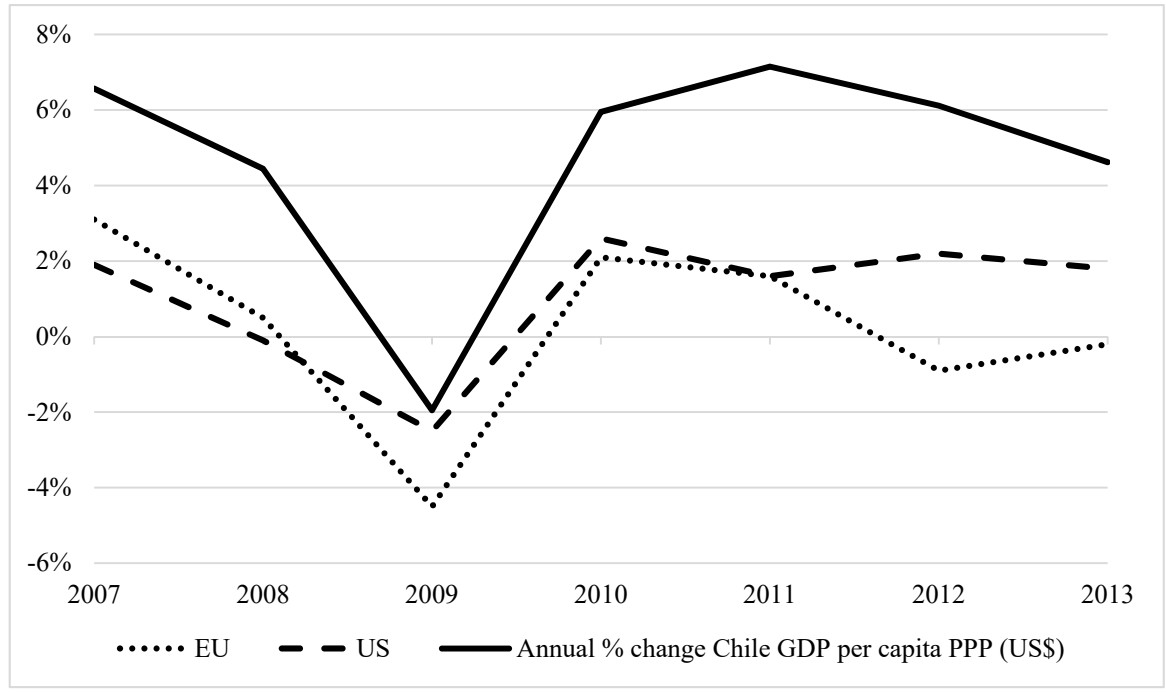

**Figure 1.** Annual percentage change in Chile, EU, and US GDP.

---

[2] Foreign equity holdings was about 33% of total AFP's equity securities during 2009–2012. See https://www.spensiones.cl/apps/boletinEstadistico/. A sixth AFP was created only in 2015.

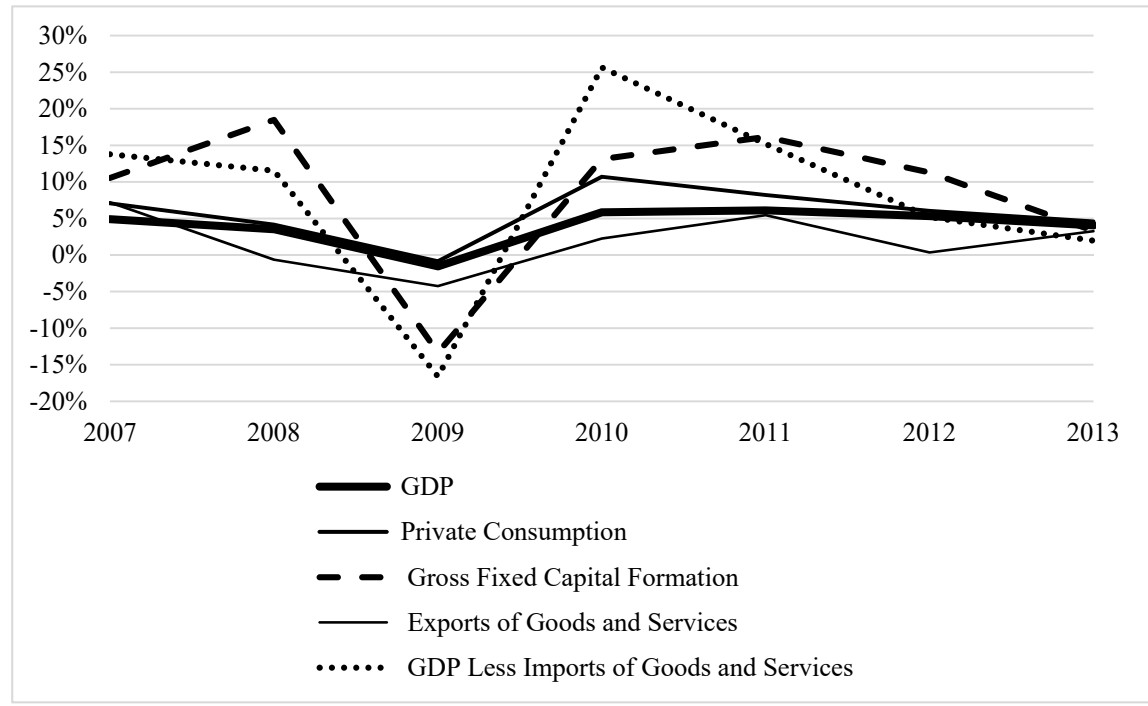

**Figure 2.** Annual percentage change in GDP by selected expenditures.

Figure 3 below charts the annual returns for the Chilean, US, and EU equity markets across the 2007–2013 study period. Returns declined in all three equity markets during the 2008 US mortgage crisis but equity market returns were more varied surrounding the EU debt crisis of 2011. Note that the decline in the Chilean equity market return associated with the EU debt crisis was large relative to the US crisis. Figure 4 displays household financial net worth, and shows that changes in the value of equity shares are closely related to changes in the value of household net worth. Chilean households experienced a twenty-percent decline in net worth during the 2008 crisis, which was comparable to the decline in the net worth of US households. But the effect on Chilean retail investors would be substantial. Hence, if changes in household net worth have real effects on spending and investment, and given that changes in household net worth are associated with market returns, then it is important to improve our understanding of the various mechanisms and policies that magnify and mitigate contagion from GFC's into the real economy. Figures 3 and 4 show that the recovery of Chilean investors is slower than that of US and EU equity investors.

A key feature of Chilean equity ownership displayed in Figure 5 below is the high concentration of shares held by insiders. Insiders are the largest shareholder group in Chile, holding over 50% of outstanding shares. Many Chilean publicly listed companies belong to a business group, and family insiders typically control the holding company (Lefort and Walker 2007). Further, highly concentrated ownership in Chilean companies has been shown to affect the liquidity of shares and their governance (Morales et al. 2013; Gjerde et al. 2013).

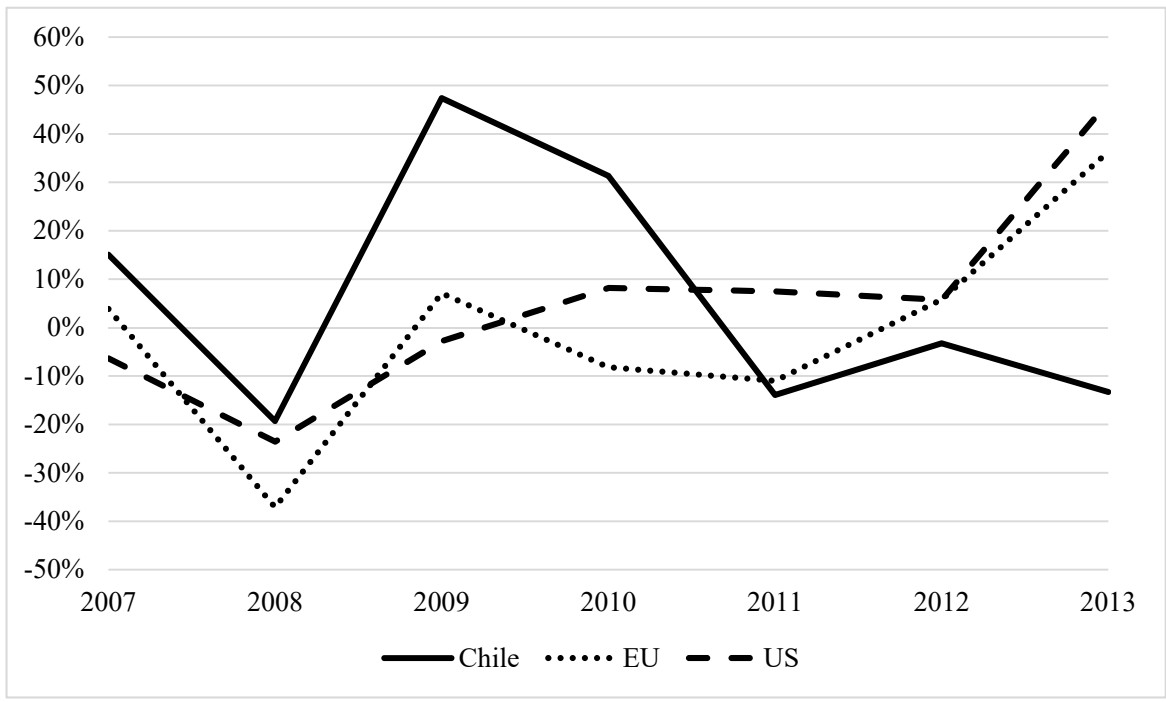

**Figure 3.** Equity market returns for Chile, EU, and US.

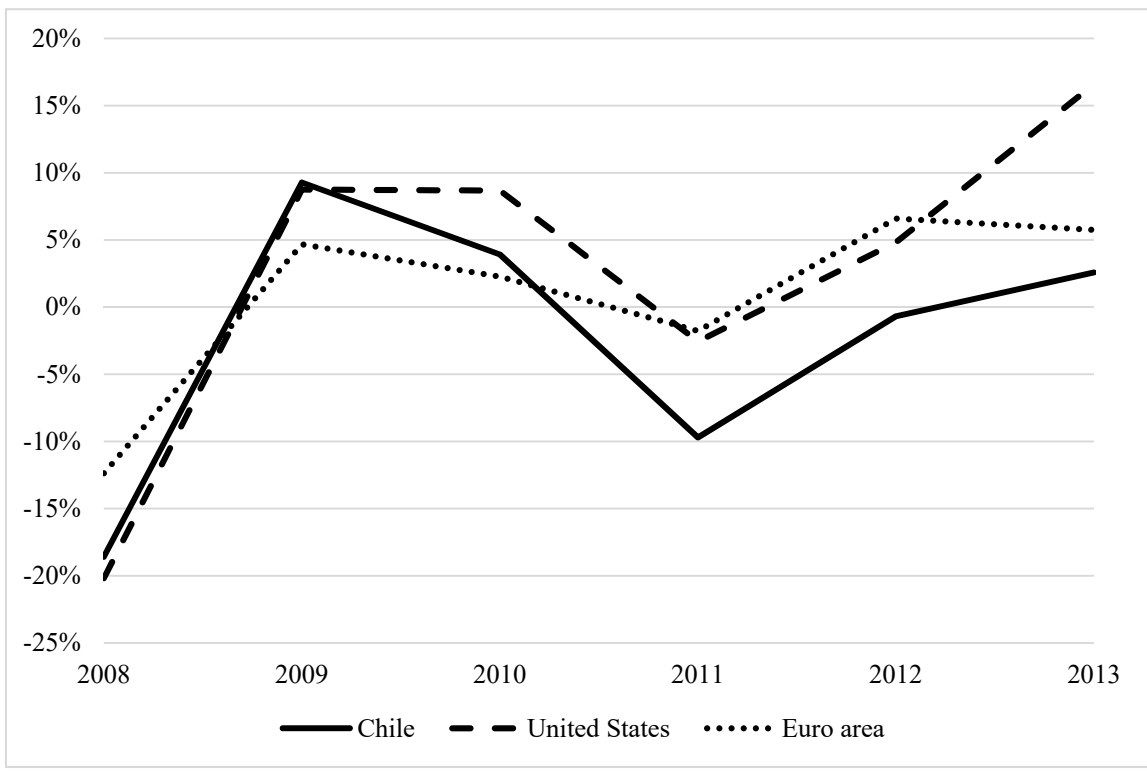

**Figure 4.** Change in households' net worth as a percentage of Gross Domestic Investment for Chile, EU, and US.

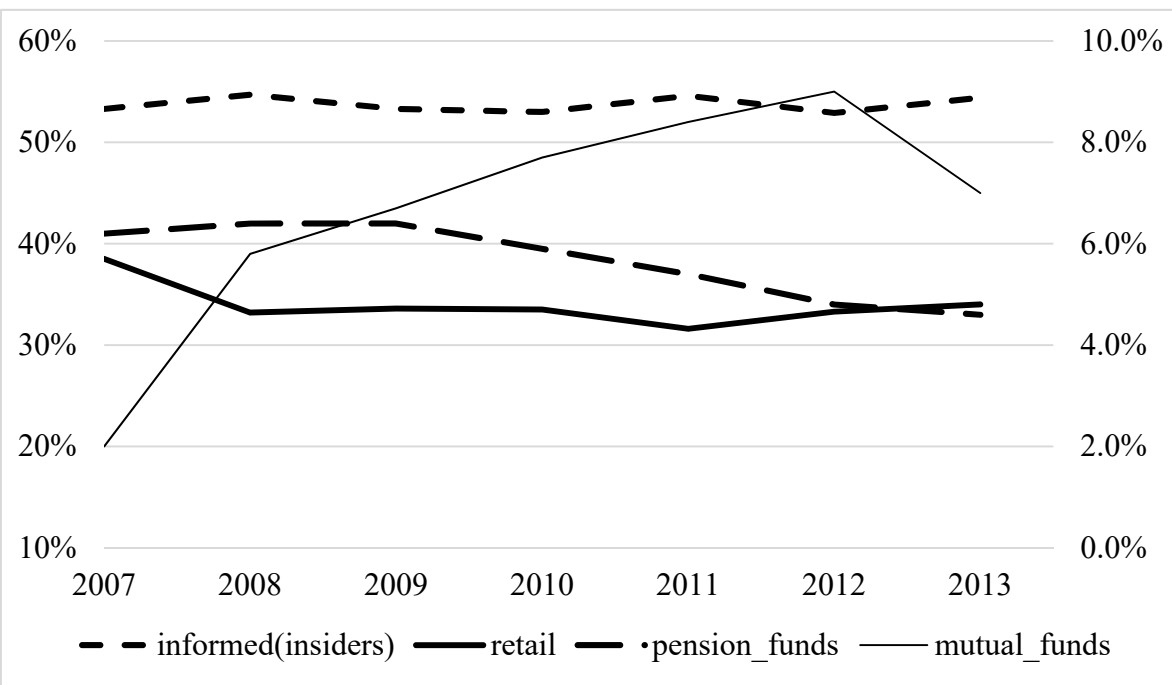

**Figure 5.** Percentage of outstanding shares held by insiders, retail, pension funds, and mutual funds. Holdings of insiders and retail is measured on the left axis, AFP's ("Adminstradoras de Fondos de Pensiones") and mutual fund ownership is measured on the right axis.

Retail investors make up a second shareholder group, controlling the outstanding shares not held by insiders and institutional investors. Inspection of Figure 5 reveals a noticeable decline in the equity holdings of retail investors during 2008 and a simultaneous increase in the equity holdings of insiders and the two groups of institutional investors. The decline in the equity holdings of retail investors in 2008 suggests that selling by retail investors may be responsible for the Chilean contagion results cited in the Dungey and Gajurel (2015) and Bekaert et al. (2014) studies.

Institutional investors are another important group of shareholders in Chile and consist of pension funds and mutual funds. Institutional investors control one in seven shares. In Figure 5, mutual fund and AFP holdings were almost equal at 6% in 2009, the year that most publicly held companies adopted IFRS and global financial markets began recovering from the US mortgage crises. Over the next four years, AFP's reduced their equity holdings by approximately a third, while mutual funds increased their equity holdings by over a third before falling back towards 7% in 2013.

AFP's have a steady inflow of funds from payroll deduction associated with Chile's privatized pension system. Given that AFP's are relatively passive portfolio managers, a potential explanation for their behavior shown in Figure 5 is that an increase in redemption orders and asset reallocation driven by two GFC's forced AFP managers to sell equity holdings at a rate that exceeded equity purchases associated with payroll contributions.

Another feature apparent in Figure 5 is that relative to the change in the holdings of insiders, the equity holdings of mutual funds and AFP's varied significantly during the study period but trended in opposite directions. The changes in insider holdings are much smaller in percentage terms, but a small change in insider holdings can have a relatively large effect given that insiders own over half of all outstanding shares on the Chilean stock market. Compared to insiders, mutual fund holdings are much smaller. Nevertheless, mutual fund holdings of Chilean equities may represent only a portion of their globally diversified portfolio, and they can have a variety of investment objectives.

Although mutual funds and AFP's may share investment goals, the regulatory burden is much less for mutual funds. The AFP's have limitations on the amount of funds they can invest in foreign securities, while mutual funds may have foreign management and substantial exposure to global

markets. It follows that changes in mutual fund holdings relative to changes in AFP holdings may be less sensitive to changes in Chilean economic conditions and more sensitive to changes in global economic conditions. However, changes in the holdings of mutual funds and AFP's in Figure 5 suggest that both AFP's and mutual funds are relatively sensitive to changes in the local market conditions, although their incentives for holding Chilean equity shares may differ. Of note is the sharp increase in shares held by mutual funds in 2008, perhaps due to mutual fund managers viewing Chile as a relatively less risky equity market (compared to other emerging markets) during the 2008 GFC. Thus, mutual fund purchases of Chilean equity shares may have mitigated the transmission of contagion from the 2008 GFC into the Chilean equity market. This understanding helps focus our literature review and hypotheses development.

## 3. Literature Review

Bekaert et al. (2014) used a three-factor asset-pricing model of country-sector equity returns to distinguish between equity-market co-movements due to a US-specific factor, a global financial factor, and a domestic factor. Accordingly, they state (Bekaert et al. 2014, p. 2602) "the inclusion of three different factors in our model enables us to distinguish between three types of contagion. Contagion may stem from the US or from the global financial sector, implying a high co-movement of domestic sector portfolios with the US or the global factors. We will label these "US contagion" and "global contagion", respectively. Alternatively, while investors may continue to discriminate across countries in response to global or US-specific shocks during crises, they may discriminate less across stocks within countries in response to idiosyncratic, country-specific shocks, thus giving rise to what we call domestic contagion". Among the authors' conclusions was the finding that domestic contagion was the major source of contagion for Chile during the 2008 GFC.

Dungey and Gajurel (2015) refer to the global contagion as systematic contagion. Systemic contagion is typically associated with a common global shock experienced simultaneously across markets that is similar to the contagion caused by the 2020 coronavirus-related financial shock. They identify a second type of contagion as idiosyncratic contagion, which occurs when a shock from a crisis originating country spreads to another economy. For example, the US-based mortgage crisis causing price movements in the Latin American stock markets due to fear of global debt defaults. They define a third type of contagion as volatility contagion, which occurs when a shock causes an increase in volatility in one market and then the effects spread to foreign markets. The authors conclude that the 2008 GFC resulted in about one-fifth of the countries studied, including Chile, experiencing the US-sourced idiosyncratic contagion. This result is consistent with Bekaert et al. (2014) given that the definition of domestic contagion in that study is consistent with the definition of idiosyncratic contagion in Dungey and Gajurel (2015). Moreover, idiosyncratic contagion was the main source of contagion for several advanced economies including Japan and France, partly because of these markets' greater level of integration with global markets relative to frontier markets like Vietnam, which was studied by Bui et al. (2018).

Ma et al. (2018) noted that during crisis periods, the unavailability of liquidity is an important channel through which market volatility affects stock returns in equity markets. These authors suggest that market makers faced with credit constraints and heightened uncertainty pull back from providing liquidity. Consequently, they argue that illiquidity drives firm value further from intrinsic value, which is thought to be a feature of the 2020 coronavirus shock. Further inspection of Figure 5 above reveals that, during our study period, the ownership level of insiders nearly mirrored that of retail investors. Insider holdings rose (fell) when the ownership of retail investors fell (rose). Thus, it is possible that insiders serve as liquidity providers, a role that takes on greater importance during a GFC given the finding in Ma et al. (2018) that market makers withdraw from providing liquidity during a crisis.

Given that we use equity holdings instead of equity returns to study contagion, we review the literature on portfolio rebalancing as potential drivers of contagion. Kodres and Pritsker (2002) show that through cross-market portfolio rebalancing, investors transmit idiosyncratic shocks from one market to another when they adjust their portfolios' exposures to shared macroeconomic risks.

Their model shows that portfolio rebalancing can generate contagion between markets that do not directly share common economic risks. Moreover, they show that contagion among emerging markets can occur indirectly through a third developed country (like the US) without significantly affecting prices in the developed country. The authors identify "correlated information channels" and "correlated liquidity shock channels" as potential mechanisms to transmit the contagion. Particularly, Kodres and Pritsker (2002, p. 770) state that, "Under the correlated information channel, price changes in one market are perceived as having implications for the values of assets in other markets that causes their prices to change as well. The correlated liquidity shock channel posits that when some market participants need to liquidate some of their assets to obtain cash, they choose to liquidate assets in a number of markets, effectively transmitting the shock to other markets (Calvo 1999)". Additionally, when portfolio rebalancing occurs in markets with information asymmetries, the resulting price co-movements are exaggerated because the order flows are misconstrued as being information-based flows. This led them to conclude that price co-movements are exaggerated in emerging markets that have a significant proportion of uninformed investors.

Boyer et al. (2006) provide evidence that stock market crises spread globally through the asset holdings of international retail investors. These studies argue that uninformed but rational investors are unable to distinguish between selling based on liquidity shocks and selling based on fundamental economic shocks in the presence of recursive relationships between a country's fiscal policies, monetary policies, and market liquidity (see Chowdhury et al. 2018). This may be the case with the 2020 coronavirus shock contagion that is rattling global capital markets.

Kyle and Xiong (2001) also consider the relationship between contagion and imperfect information. The authors separated investors into three groups, noise traders trading randomly in only one market, long-term value-based investors trading on fundamentals and providing liquidity, and convergence traders who trade optimally in multiple markets. Convergence traders are perfect competitors, and rational in the sense that their trading strategies correctly take into account the effect of all market participants on the price dynamics in more than one market. Thus, convergence traders, unlike long-term investors, aggressively exploit short-term opportunities by taking the other side of noise trading. On the other hand, long-term investors are not fully rational in the sense that they tend to ignore the short-term opportunities caused by noise traders. Moreover, Kyle and Xiong (2001) argue that it is possible for contagion to result from confused convergent traders if convergence traders cannot distinguish between liquidity shocks and shocks to company fundamentals.

Gromb and Vayanos (2010) refer to the risk that stems from noise traders as non-fundamental risk to emphasize the possibility that demand shocks may be unrelated to asset payoffs, which can arise from rational short-term trading behavior by convergence traders. A consequence is that price declines are exaggerated when arbitrageurs, such as convergence traders, are forced to liquidate their position and long-term investors like insiders fail to provide liquidity to the market. Thus, if convergence traders are arbitrageurs who cannot exit their positions in an illiquid market during a crisis, then they can have a destabilizing effect on emerging markets that can lead to a market crash.

## 4. Methodology

The analysis attempts to measure the amount of variation in equity holdings explained by lagged equity holdings, spillover, contagion, and control variables during the 2007–2013 study period. Total outstanding shares in the Chilean equity market are allocated among four groups of investors according to ownership data obtained from the Superintendcia Valores y Seguros that is referred to as the SVS, which is the equivalent of the United States Securities and Exchange Commission in Chile. The four ownership groups are pension funds, mutual funds, retail investors, and insiders.

A goal of modeling equity holdings instead of equity returns is to determine if any of the four investor groups made substantial changes to their equity holdings in response to the two GFC's within the study period. If they did, then we can conclude that portfolio rebalancing may have served as a mechanism to transmit contagion into the domestic economy. Furthermore, with our methodology,

it may be possible to identify which investor groups magnified (or mitigated) contagion effects from the GFC's.

The spillover market's proxies are the annual return on three foreign equity indices consisting of an MSCI US equity index (referred to as "US Spillover"), an MSCI EU equity index (referred to as "EU Spillover"), and an MSCI global equity index (referred to as "Global Spillover")[3]. The annual return on the Chilean equity market measures domestic market influence on equity holdings. As a first step, the domestic (Chilean) equity return is orthogonalized by regressing it against the other three returns. Hence, the residuals represent the portion of variation in the domestic index returns that is not explained by variation in the other three external indices. The residuals become the domestic factors within the regression model. Dungey and Gajurel (2015) and Bekaert et al. (2014) employ a similar methodology to measure spillover effects on the equity returns in their studies.

Contagion is defined as the additional or unexpected spillover that may occur during a GFC. We applied the methodology employed by Dungey and Gajurel (2015) and Bekaert et al. (2014) to create "crisis" dummies intended to capture the additional spillover, i.e., contagion, which occurs during a GFC. We identified 2008 as the crisis year associated with the US mortgage crisis, and the year 2011 as the crisis year associated with the EU debt crisis. Each GFC dummy variables equals zero in non-crisis years. In 2008, the GFC dummy "Global_crisis_08" equals the 2008 annual return for one of the three equity market indices, and similarly for the 2011 GFC dummy "Global_crisis_11". We estimated a separate model for each of the three foreign spillover markets and a fourth model with no spillover market. Each model contains both GFC dummies and the model without a spillover market isolates the impact of variation in the orthogonalized domestic index return on equity holdings.

The relationship between equity holdings and several of the modeled control variables are also of interest. For example, we controlled for the impact of long-term debt (Percent_bonds) on equity holdings and interpreted the results in terms of creditor monitoring that is a proxy for external corporate governance. The liquidity control variables include the annual number of trading days per company with no change in return (Zero_return_days), and the effective spread (Effective_spread). Firms operating in the financial sector are identified by a dummy variable (Financial_sector) equal to "1" for financial sector firms and "0" otherwise. Note that Dungey and Gajurel (2015) restrict their analysis to firms in the financial sector based on the observation that the banking sector represent a channel through which contagion may be transmitted across borders. Chilean firms adopted IFRS and fair value accounting standards during the study period, so we included a control variable based on the year of adoption, and we expect the requirements for fair market valuation requirements in IFRS to significantly affect the financial sector. Many firms voluntarily adopted fair value standards early, and we controlled for early adoption through a dummy variable (Fair_Value) that equals "1" upon adoption and each year thereafter, and 0 otherwise.

Dungey and Gajurel (2015) and Bekaert et al. (2014) both address the impact of portfolio return volatility in their multi-country studies of contagion. Bekaert et al. (2014) implicitly control for volatility by introducing a volatility ratio of market volatility to factor volatility. Dungey and Gajurel (2015) explicitly model volatility contagion and find that while volatility spillover is common to most countries, roughly 40% of them experience volatility-driven contagion. Furthermore, they note that volatility contagion is rarely the only driver of contagion effects and is associated with policy uncertainty too. Given that the focus of the current study is on a single country, we modeled variation in equity holdings instead of equity returns, and we defined the average standard deviation of daily returns per company as our control for "volatility".

Portfolio turnover and investor herding are measured by the inclusion of lagged dependent variables on the right-hand side each model. For any given shareholder group, we interpreted a

---

[3]   MSCI Inc., is an American Finance Company headquartered in New York City, and it is a global provider of equity, fixed income, and stock market indexes and multi-asset portfolio analysis tools. Its URL is: https://www.msci.com/.

statistically significant parameter estimate on lagged equity holdings as a measure of portfolio turnover. As an extreme example, if equity holdings do not change form one period to another, then this period's equity holdings equal last period's equity holdings, so that the parameter estimate is 1 and the lagged holdings can explain all the variation in current period equity holdings. Similarly, herding is defined for shareholder group "i" as a statistically significant parameter estimate on the lagged holdings of shareholder group "j". In other words, modeled herding occurs when the trades of one investor group are highly correlated with the lagged trades of a second investor group[4].

The equity holdings of AFP's, mutual funds, and insiders are modeled in Equation (1) as a function of the two institutional investors and insiders. The three groups of investors are superscripted with "i", "j", and "k" in Equation (1) below. Thus, for each investor group i, and given two other investor groups, j and k, the general holdings model is expressed in Equation (1) as:

$$
\begin{aligned}
h^i = {} & \beta_0 + \beta_1 \, Lag\_h^i + \beta_2 \, Lag\_h^j + \beta_3 \, Lag\_h^k + \beta_4 \, Effective\_spread + \beta_5 \, Zero\_return\_days \\
& + \beta_6 \, Percentage\_bonds + \beta_7 \, Fair\_value + \beta_8 \, Volatility + \beta_9 \, Financial\_sector \\
& + \beta_{10} \, Domestic\_contagion + \beta_{11} \, Spillover\_market + \beta_{12} \, Global\_crisis\_08 \\
& + \beta_{11} \, Global\_crisis\_11 + \varepsilon^i.
\end{aligned}
$$

The retail investor model is identical to Equation (1), but excludes lagged pension fund and mutual fund holdings[5].

### 4.1. Hypotheses

Our models separate the equity market response to the two GFC's into changes in the equity holdings of four distinct shareholder groups: insiders, local pension funds or AFP's, mutual funds, and retail investors. In this context, to provide directional hypotheses, we state our hypotheses in terms of the type of investor whose holdings are likely to be a mechanism for transmitting the contagion during a GFC, and those that are not likely to be mechanisms for transmitting the contagion.

**Hypothesis 1 (H1).** *Change in the level of local pension fund holdings is not a mechanism for the transmission of GFC effects to the Chilean financial markets.*

**Hypothesis 2 (H2).** *Change in the level of mutual fund holdings is not a mechanism for the transmission of GFC effects to the Chilean financial markets.*

**Hypothesis 3 (H3).** *Change in the level of retail investor holdings is a mechanism for the transmission of GFC effects to the Chilean financial markets.*

**Hypothesis 4 (H4).** *Change in the level of insider holdings is not a mechanism for the transmission of GFC effects to the Chilean financial markets.*

To test the hypotheses, four models were estimated for insiders, mutual funds, pension funds, and retail investors that complement the studies of Dungey and Gajurel (2015) and Bekaert et al. (2014). However, as shown above, Equation (1) replaces their dependent variable based on equity returns with the equity holdings of insiders, mutual funds, pension funds, and retail investors. An analysis based on equity holdings facilitates an examination of the possibility that institutional managers, retail investors,

---

[4] Durbin-h test indicates the presence of autocorrelation in all models except the insider-holding model. Estimation of the models with auto regression (AR1) or ordinary least square (OLS) does not have a qualitatively different impact on the results in Tables 3–6. Further, all continuous variables are standardized with mean of 0 and standard deviation of 1.

[5] We do not include all four investor groups in a single model because retail investor holdings equal the residual of total outstanding shares minus the holdings of the other three investor groups. Thus, the holdings of any single group are a linear combination of the other 3. The current specification provides the best fit to the data, and facilitates a discussion of the relationship observed in Figure 5 between retail investors and insiders.

and insiders provide liquidity when market makers pull back from the provision of providing liquidity during GFC's.

In summary, AFP managers are relatively passive investors given their mandate to invest the payroll contributions of Chilean civilians. Mutual funds may have more exposure to global markets and may have viewed Chile as a relatively safe haven during the 2008 and 2011 GFC's. Retail investors are perhaps the least informed group of Chilean investors and this group is most likely to misinterpret the impact of foreign shocks on domestic fundamentals, and may also be prone to herd behavior during panics. Insiders are informed traders with superior information on the value of the firm, and the assumption is that they do not panic during a GFC. As such, insiders should view GFC-driven panic selling as buying opportunities and add liquidity to the emerging market during sell-offs.

The tradeoff between risk reduction associated with creditor monitoring and increased risk of default associated with two GFC's during the study period motivates a fifth hypothesis. Hypothesis 5 tests for the possibility that a higher level of publicly issued debt in the capital structure may motivate investors to increase equity holdings to take advantage of credit monitoring. The converse of this hypothesis is that investors held fewer shares in companies with more publicly issued debt to avoid exposure to a heightened probability of default that generally existed during the study period. The tradeoffs are particularly interesting in light of the fact that the Chilean banking model is a traditional deposit-loan model rather than an investment banking/securitization model, which is also the case in most emerging markets. During normal times, debt is a significant source of funds for Chilean firms. Total debt represents just over 45% of our sample firms' capital structure, while long-term debt represents nearly 25% of our sample firms' capital structure. Publicly issued long-term bonds represent close to 12% of the capital structure. The relatively high level of debt in the capital structure of Chilean companies is explained in part by a high concentration of shares held by insiders who seek to maintain majority control by issuing debt instead of equity shares (Lefort and Walker 2007). In emerging markets, Denis and McConnell (2003) note the importance of the ability of creditors to monitor insiders. Additionally, studies have shown that creditor monitoring is present except when a country has an easy monetary policy (Lopez-de-Foronda et al. 2018). Thus, by including the percentage of bond holdings in our models, we allow for the conflicting effects of it on equity holdings, a heightened probability of default during the study period, potential value of creditor monitoring, and the possibility of flight to quality (Optiz and Szimayer 2018). Moreover, we allow the data to speak on the direction of the correlation of this variable with different equity holdings. Thus, we hypothesize that equity investors recognize a relationship between equity value and the level of debt in the capital structure during a crisis.

**Hypothesis 5 (H5).** *The equity holdings of mutual funds, local pension funds, and retail investors are sensitive to the presence of more debt in the capital structure.*

*4.2. Sample*

For hypothesis testing, we used price and volume data provided by the regulators of the Santiago Stock Exchange, for the period January 2007 through December 2013. Equity holdings were obtained from the Superintendencia de Valores y Seguros (SVS) that is also referred to as the Commission for the Financial Market. Additionally, we collected annual measures of financial statement items from the Economatica database. Annual equity market returns were obtained from MSCI for Chile, US, EU, and global stock market indices. Annual equity returns were derived using local currencies. Real GDP growth rates and components for Chile, EU, and US were obtained from the Statistics Database Banco Central De Chile. Total portfolio investment was from the International Monetary Fund Coordinated Portfolio Investment Survey (CPIS—Tables 12 and 13). Household net worth was from the Organization for Economic Corporation and Development (OECD) Stat, household dashboard. Further, all firms were required to adopt fair value measurement standards by December

2013, as prescribed by International Accounting Standards Board[6]. The final sample includes 63 firms, which we pooled across the period 2007 through 2013, producing a total of 339-pooled firm years after removing observations with missing data. Appendix A lists the variables and their definitions.

## 5. Results

Table 1 provides descriptive statistics that are based on 339 firm years. The high degree of insider ownership in Chile is evidenced by an average insider ownership level of 53.12%. In contrast, the average local pension fund holdings were 5.88%, and mutual funds held 7.8% of the outstanding shares. Retail investors, who control 33.2% of shares, are the second largest shareholders.

**Table 1.** Mean, standard deviation (SD), minimum (min), and maximum (max) values for the sample.

| Variable | Mean | SD | Min | Max |
|---|---|---|---|---|
| Insiders | 53.12% | 21.30% | 0.00% | 99.28% |
| Pension funds | 5.88% | 6.30% | 0.00% | 25.11% |
| Mutual funds | 7.80% | 9.88% | 0.00% | 39.00% |
| Retail | 33.20% | 23.65% | 0.00% | 97.09% |
| Lag_retail | 34.32% | 23.22% | 0.00% | 97.09% |
| Lag_Insiders | 52.60% | 20.70% | 0.00% | 97.80% |
| Lag_Pension funds | 6.20% | 6.64% | 0.00% | 25.11% |
| Lag_Mutual funds | 6.89% | 9.58% | 0.00% | 42.49% |
| Effective_spread | 1.52 | 1.72 | 0.18 | 14.41 |
| Zero_return_days | 11.28 | 8.41 | 1 | 43 |
| Percent_bonds | 11.96% | 16.14% | 0.00% | 83.52% |
| Volatility | 2.67 | 1.83 | 0.06 | 15.34 |
| Financial_sector | 0.31 | 0.46 | 0 | 1 |
| Fair_value | 30.68% | 46.18% | 0.00% | 100.00% |
| Domestic_contagion | 0.00% | 8.67% | −10.40% | 11.61% |
| Global spillover | 3.30% | 12.84% | −16.86% | 24.06% |
| US spillover | 5.66% | 11.88% | −12.98% | 27.46% |
| EU spillover | 1.03% | 16.58% | −25.39% | 23.48% |

Table 2 displays correlations for the study variables. Insider ownership is significant and negatively correlated with both contemporaneous and lagged values of pension fund and mutual fund holdings, at $p < 0.01$ and $p < 0.10$ levels. As expected, insiders' ownership is not correlated with any of the external spillover proxies or the domestic contagion factor, EU Spillover, Global Spillover, and US Spillover. Further, insiders' ownership is positively related to the liquidity proxy effective_spread at the $p < 0.05$ level. This is consistent with the idea that higher levels of insider ownership suggest less liquidity and wider effective spreads. The creditor monitoring proxy Percent_bonds is positively and significantly correlated with the holdings of Insiders, Pension_funds, Lag_insiders, Lag_Pension_funds, and Lag_mutual_funds at the $p < 0.01$ level. It is also significant and negatively correlated with holdings of "Retail" investors at the $p < 0.01$ level. Fair_value is positively correlated with Pension_funds and Mutual_funds at the $p < 0.10$ l and $p < 0.01$ levels, but negative and significantly correlated with Retail, Zero_ret_days, and the Financial_sector dummy at the $p < 0.05$ level. These results suggest that retail investor behavior is different from the other groups, and the financial sector is less likely to have adopted fair value reporting of assets and liabilities.

---

[6] All but 7 firms adopted IFRS in 2009, so we omitted those 7 firms. Early adopters of IFRS 13 provided levels 1–3 fair value measurements in their financial statements, thus, they had recognized (not simply disclosed) fair value measurement before the mandated date of December 2013.

**Table 2.** Pearson correlation matrix.

| | Variable | 1 | 2 | 3 | 4 | 5 | 6 | 7 | 8 | 9 | 10 | 11 | 12 | 13 | 14 | 15 | 16 | 17 |
|---|---|---|---|---|---|---|---|---|---|---|---|---|---|---|---|---|---|---|
| 1. | 1. Insiders | 1 | | | | | | | | | | | | | | | | |
| 2. | 2. Pension funds | −0.164 *** | 1 | | | | | | | | | | | | | | | |
| 3. | 3. Mutual funds | −0.102 * | 0.448 | 1 | | | | | | | | | | | | | | |
| 4. | 4. Retail | −0.814 *** | −0.305 | −0.444 | 1 | | | | | | | | | | | | | |
| 5. | 5. Lag_retail | −0.738 *** | −0.288 | −0.395 | 0.906 *** | 1 | | | | | | | | | | | | |
| 6. | 6. Lag_Insiders | 0.918 | −0.146 *** | −0.066 | −0.761 *** | −0.818 *** | 1 | | | | | | | | | | | |
| 7. | 7. Lag_Pension funds | −0.141 *** | 0.922 *** | 0.406 | −0.288 | −0.297 | −0.154 *** | 1 | | | | | | | | | | |
| 8. | 8. Lag_Mutual funds | −0.098 * | 0.376 | 0.820 *** | −0.354 | −0.451 | −0.071 | 0.361 | 1 | | | | | | | | | |
| 9. | 9. Effective_spread | 0.138 ** | −0.268 | −0.435 | 0.128 ** | 0.131 ** | 0.123 ** | −0.261 | −0.405 | 1 | | | | | | | | |
| 10. | 10. Zero_return_days | −0.102 * | −0.131 ** | −0.255 | 0.233 | 0.250 | −0.126 ** | −0.133 ** | −0.240 | −0.062 | 1 | | | | | | | |
| 11. | 11. Percent_bonds | 0.150 *** | 0.307 *** | 0.281 *** | −0.334 *** | −0.289 *** | 0.145 *** | 0.286 *** | 0.188 *** | −0.250 | 0.055 | 1 | | | | | | |
| 12. | 12. Volatility | 0.005 | −0.155 *** | −0.269 | 0.149 *** | 0.168 *** | −0.012 | −0.146 *** | −0.281 | 0.440 | −0.176 *** | −0.226 | 1 | | | | | |
| 13. | 13. Financial_sector | 0.145 *** | −0.426 | −0.313 | 0.113 ** | 0.089 | 0.148 | −0.448 | −0.225 | 0.087 | 0.168 *** | −0.119 ** | 0.019 | 1 | | | | |
| 14. | 14. Fair_value | 0.088 | 0.095 * | 0.196 *** | −0.187 *** | −0.238 | 0.111 | 0.135 ** | 0.242 | −0.264 | −0.135 ** | 0.105 * | −0.233 | −0.113 ** | 1 | | | |
| 15. | 15. Domestic_contagion | 0.102 | 0.037 | 0.002 | 0.000 | −0.023 | 0.008 | 0.038 | 0.012 | −0.083 | −0.166 *** | 0.037 | −0.006 | 0.015 | 0.112 ** | 1 | | |
| 16. | 16. Global spillover | −0.021 | −0.055 | 0.038 | 0.018 | −0.060 | −0.010 | −0.040 | 0.198 *** | −0.192 *** | −0.043 | −0.066 | −0.138** | 0.003 | 0.202 *** | 0.00 | 1 | |
| 17. | 17. US spillover | −0.024 | −0.067 | 0.044 | 0.021 | −0.053 | −0.022 | −0.048 | 0.212 | −0.214 | −0.094 * | −0.084 | −0.135** | 0.008 | 0.223 | 0.00 | 0.974 *** | 1 |
| 18. | 18. EU spillover | −0.021 | −0.050 | 0.038 | 0.016 | −0.064 | −0.003 | −0.036 | 0.190 *** | −0.174 *** | −0.010 | −0.053 | −0.123** | 0.001 | 0.188 *** | 0.00 | 0.983 *** | 0.924 *** |

In all the tables, *** Significant at the 1% level, ** significant at the 5% level, and * significant at 10%.

Note that Domestic_contagion in column 15 of Table 2 is uncorrelated with the spillover proxies EU Spillover, Global Spillover, and US Spillover. This is the result of orthogonalizing Domestic_contagion against the spillover proxies. However, the spillover proxies are highly correlated with each other, so we avoided collinearity issues within regressions by running a separate regression for each spillover market in addition to a baseline regression with no spillover markets. The outcome of this estimation strategy is that it allows Domestic_contagion to appear in all regressions to isolate the impact of the orthogonalized domestic market effect after controlling for external market effects.

Regression results for the equity holdings of AFP's, mutual funds, retail investors, and insiders are summarized in Tables 3–6, respectively. Each table contains four models with each model identified by the variable "Spillover_market". Model 1 has no spillover market and serves as the baseline model. Spillover_Market equals EU Spillover in Model 2, Global Spillover in Model 3, and US Spillover in Model 4. Recall from our discussion that spillover refers to the expected cross-border co-movement in asset prices, while the term "contagion" refers to unexpected or excessive spillover. With these definitions in mind, the variables "Global_crisis_08" and "Global_crisis_11" measure contagion effects associated with the US mortgage crisis in 2008 and the EU debt crisis in 2011. Each of those variables is "0" during non-crisis years and take on the value of the spillover variable in crisis years. The same methodology was employed by Bekaert et al. (2014) and Dungey and Gajurel (2015) to measure contagion effects.

**Table 3.** Regression AR1 coefficient estimates of pension fund holdings on contagion models for the 2008 and 2011 global financial crises (*t*-stats).

| Variable | Model (1) | Model (2) | Model (3) | Model (4) |
|---|---|---|---|---|
| Intercept | 0.019 | −0.056 | −0.035 | −0.011 |
| | (0.307) | (−1.149) | (−0.742) | (−0.249) |
| Lag_insiders | −0.007 | −0.012 | −0.009 | −0.008 |
| | (−0.298) | (−0.486) | (−0.377) | (−0.309) |
| Lag_pension_funds | 0.856 *** | 0.851 *** | 0.854 *** | 0.856 *** |
| | (30.808) | (30.621) | (30.737) | (30.774) |
| Lag_mutual_funds | 0.051 * | 0.048 * | 0.049 * | 0.050 * |
| | (1.888) | (1.794) | (1.831) | (1.861) |
| Zero_return_days | −0.020 | −0.032 | −0.024 | −0.020 |
| | (−0.831) | (−1.286) | (−0.992) | (−0.818) |
| Effective_spread | −0.038 | −0.045 * | −0.041 | −0.038 |
| | (−1.462) | (−1.709) | (−1.557) | (−1.465) |
| Percent_bonds | 0.036 | 0.043 * | 0.043 * | 0.041 |
| | (1.475) | (1.770) | (1.738) | (1.645) |
| Volatility | −0.003 | 0.002 | 0.002 | 0.001 |
| | (−0.121) | (0.070) | (0.086) | (0.034) |
| Fair_value | −0.079 | −0.066 | −0.073 | −0.077 |
| | (−1.461) | (−1.213) | (−1.344) | (−1.430) |
| Financial_sector | −0.043 | −0.040 | −0.041 | −0.042 |
| | (−0.750) | (−0.700) | (−0.729) | (−0.747) |
| Domestic_contagion | 0.035 | 0.064 ** | 0.055 * | 0.048 * |
| | (1.351) | (2.236) | (1.950) | (1.707) |
| Global_crisis_08 | −0.028 *** | −0.018 *** | −0.022 *** | −0.021 ** |
| | (−2.613) | (−3.233) | (−2.853) | (−2.483) |
| Global_crisis_11 | −0.0036 | −0.0062 | −0.0076 | −0.0039 |
| | (−0.459) | (−0.843) | (−0.528) | (−0.107) |
| Spillover_market | | 0.103 ** | 0.075 * | 0.043 |
| | | (2.268) | (1.760) | (1.174) |
| Adj. $R^2$ | 0.832 | 0.832 | 0.832 | 0.832 |

In all the tables, *** Significant at the 1% level, ** significant at the 5% level, and * significant at 10%.

**Table 4.** Regression AR1 coefficient estimates of mutual fund holdings on contagion models for the 2008 and 2011 global financial crises (*t*-stats).

| Variable | Model (1) | Model (2) | Model (3) | Model (4) |
|---|---|---|---|---|
| Intercept | 0.010 | 0.005 | 0.041 | 0.051 |
| | (0.186) | (0.074) | (0.650) | (0.876) |
| Lag_insiders | −0.010 | −0.010 | −0.008 | −0.009 |
| | (−0.287) | (−0.295) | (−0.249) | (−0.274) |
| Lag_pension_funds | 0.022 | 0.022 | 0.023 | 0.021 |
| | (0.587) | (0.580) | (0.602) | (0.571) |
| Lag_mutual_funds | 0.682 *** | 0.681 *** | 0.683 *** | 0.682 *** |
| | (18.624) | (18.577) | (18.638) | (18.681) |
| Zero_return_days | −0.110 *** | −0.110 *** | −0.107 *** | −0.110 *** |
| | (−3.325) | (−3.275) | (−3.231) | (−3.349) |
| Effective_spread | −0.152 *** | −0.153 *** | −0.151 *** | −0.153 *** |
| | (−4.256) | (−4.240) | (−4.212) | (−4.271) |
| Percent_bonds | 0.102 *** | 0.103 *** | 0.098 *** | 0.094 *** |
| | (3.102) | (3.083) | (2.920) | (2.812) |
| Volatility | −0.010 | −0.010 | −0.014 | −0.017 |
| | (−0.308) | (−0.297) | (−0.404) | (−0.505) |
| Fair_value | −0.032 | −0.031 | −0.036 | −0.035 |
| | (−0.436) | (−0.421) | (−0.492) | (−0.478) |
| Financial_sector | −0.235 *** | −0.235 *** | −0.236 *** | −0.235 *** |
| | (−3.073) | (−3.064) | (−3.086) | (−3.087) |
| Domestic_contagion | 0.014 | 0.016 | 0.001 | −0.010 |
| | (0.405) | (0.412) | (0.012) | (−0.266) |
| Global_crisis_08 | −0.060 *** | −0.015 ** | −0.014 | −0.014 |
| | (−4.134) | (−2.036) | (−1.376) | (−1.165) |
| Global_crisis_11 | 0.006 | −0.006 | −0.001 | 0.005 |
| | (0.574) | (−0.542) | (−0.032) | (0.105) |
| Spillover_market | | 0.007 | −0.051 | −0.078 |
| | | (0.114) | (−0.865) | (−1.558) |
| Adj. $R^2$ | 0.696 | 0.695 | 0.697 | 0.698 |

In all the tables, *** Significant at the 1% level, ** significant at the 5% level.

Hypothesis H1 states that change in the level of pension fund or AFP holdings is not a mechanism for the transmission of GFC effects into the Chilean financial market. The results in Table 3 below contain no evidence to reject H1. The parameters on the contagion variable Global_crisis_08 are negative and significant at $p < 0.05$ across all spillover markets (i.e., β = −0.028 for the baseline Model 1, β = −0.018 for the EU Model 2, β = −0.022 for the Global Model 3, and β = −0.021 for the US Model 4). Additionally, the only spillover market with a *p*-value below 0.05 is the EU market with a positive relationship to AFP equity holdings. An interpretation is that the equity holdings of AFP's are positively correlated with EU equity market returns during normal times and increase along with increases in the return to EU equity. However, during the 2008 GFC, the equity holdings of AFP's increased as the return in all foreign equity markets declined. Thus, AFP's do not appear to be a source of the transmission for the contagion Chile experienced during the 2008 GFC as identified by Bekaert et al. (2014) and Dungey and Gajurel (2015). Furthermore, the lack of significance between the equity holdings of AFP's and the crisis variable "Global_crisis_11" indicates that pension fund holdings were largely unaffected by the 2011 GFC. Therefore, our results support hypothesis H1 that AFP's were not a source of transmission of the 2008 and 2011 crises.

Hypothesis H2 states that change in the level of mutual fund holdings is not a mechanism for the transmission of GFC effects. Table 4 below shows that none of the Spillover_market effects are significant. The measure of contagion, Global_crisis_08, is negative and significant for the Chilean market (β = −0.060, $p < 0.01$ in Model 1) and the EU market (β = −0.015, $p < 0.05$ in Model 2).

These results suggest that the equity holdings of mutual funds increased as the returns on the Chilean equity market and the EU equity market declined during the 2008 GFC. Thus, mutual funds were not a mechanism for the transmission of contagion during the 2008 GFC. Additionally, the Global_crisis_11 estimates are not significant across all markets. Further, the lack of significance on the estimate for Domestic_contagion implies there is no evidence of a domestic effect on mutual fund holdings either. Therefore, we accept hypothesis H2: the holdings of mutual funds did not serve as a transmission mechanism for contagion during the 2008 GFC.

**Table 5.** Regression AR1 coefficient estimates of retail investor holdings on contagion models for the 2008 and 2011 global financial crises (*t*-stats).

| Variable | Model (1) | Model (2) | Model (3) | Model (4) |
|---|---|---|---|---|
| Intercept | 0.005 | 0.011 | 0.013 | 0.018 |
|  | (0.119) | (0.213) | (0.267) | −0.264 |
| Lag_insiders | −0.136 *** | −0.136 *** | −0.136 *** | −0.136 *** |
|  | (−2.836) | (−2.835) | (−2.835) | (−2.833) |
| Lag_retail | 0.751 *** | 0.751 *** | 0.751 *** | 0.751 *** |
|  | (14.963) | (14.899) | (14.928) | (14.943) |
| Zero_ret_days | 0.043 * | 0.044 * | 0.044 * | 0.043 * |
|  | (1.691) | (1.694) | (1.707) | −1.685 |
| Effective_spread | 0.053 * | 0.053 * | 0.053 * | 0.052 * |
|  | (1.876) | (1.882) | (1.883) | −1.871 |
| Percent_bonds | −0.083 *** | −0.083 *** | −0.083 *** | −0.083 *** |
|  | (−3.293) | (−3.285) | (−3.293) | (−3.300) |
| Volatility | −0.004 | −0.004 | −0.005 | −0.005 |
|  | (−0.148) | (−0.161) | (−0.177) | (−0.187) |
| Fair_value | 0.035 | 0.034 | 0.034 | 0.034 |
|  | (0.632) | (0.609) | (0.612) | −0.622 |
| Financial_sector | 0.100 * | 0.099 * | 0.100 * | 0.100 * |
|  | (1.808) | (1.799) | (1.802) | −1.807 |
| Domestic_contagion | −0.002 | −0.005 | −0.006 | −0.006 |
|  | (−0.069) | (−0.152) | (−0.188) | (−0.210) |
| Global_crisis_08 | 0.037 *** | 0.010 * | 0.015 * | 0.020 ** |
|  | (3.252) | (1.685) | (1.895) | −2.147 |
| Global_crisis_11 | −0.005 | 0.005 | 0.011 | −0.027 |
|  | (−0.595) | (0.595) | (0.653) | (−0.668) |
| Spillover_market |  | −0.010 | −0.014 | −0.014 |
|  |  | (−0.189) | (−0.286) | (−0.333) |
| Adj. $R^2$ | 0.827 | 0.828 | 0.828 | 0.827 |

In all the tables, *** Significant at the 1% level, ** significant at the 5% level, and * significant at 10%.

Hypothesis H3 states that change in the level of retail investor holding is a mechanism for the transmission of GFC effects. Table 5 below shows that in contrast to the negative sign on the set of crisis dummies for mutual fund and pension fund holdings, the signs on the parameter estimates for Global_crisis_08 are positive and significant for the Chilean market ($\beta = 0.037$, $p < 0.01$ in Model 1) and the US market ($\beta = 0.020$, $p < 0.05$ in Model 4). Significance is marginal for the relationship with the EU and Global equity markets in Models 2 and 3, with $p < 0.10$. These results indicate that lower equity returns in the local Chilean market and the US market during the 2008 GFC are associated with lower retail investor holdings, and allowing for marginal significance on the EU and global markets, the result holds for all three foreign markets. Hence, declines in the equity holdings of retail investors during the 2008 GFC may have been a source of transmission of the domestic contagion in Chile as identified by Bekaert et al. (2014) and the idiosyncratic contagion in Chile identified by Dungey and Gajurel (2015). The lack of significance between the equity holdings of retail investors and the proxy for the Global_crisis_11 contagion suggests that retail investors did not facilitate the transmission of the contagion during the 2011 GFC. Since we identified retail investors as a source of

transmission for the contagion associated with the 2008 crisis, but not the 2011 GFC, we conclude that there is mixed evidence to support hypothesis H3.

Hypothesis H4 states that change in the level of insider holdings is not a mechanism for the transmission of GFC effects to the Chilean financial markets. Table 6 below show that across all models, there is no significant evidence for insiders being affected by the Chilean equity market or the foreign markets. In fact, the only parameter estimate of significance is on lagged insider holdings ($\beta = 0.904$, $p < 0.01$) across all the models. It appears that the trades of insiders are driven by variables outside the scope of the study. Given this result and the failure to find any significant estimates on the crisis dummies, we conclude that there is support for hypothesis H4, suggesting that insiders do not change their equity holdings during crisis periods. Hence, as suspected, insiders may be failing to provide liquidity during a GFC that they normally do during a non-crisis period.

**Table 6.** Regression AR1 coefficient estimates of insider holdings on contagion models for the 2008 and 2011 global financial crises (*t*-stats).

| Variable | Model (1) | Model (2) | Model (3) | Model (4) |
|---|---|---|---|---|
| Intercept | −0.012 | 0.003 | −0.023 | −0.034 |
| | (−0.321) | (0.070) | (−0.478) | (−0.769) |
| Lag_insiders | 0.904 *** | 0.905 *** | 0.904 *** | 0.904 *** |
| | (37.245) | (37.073) | (37.188) | (37.281) |
| Lag_pension_funds | 0.014 | 0.015 | 0.014 | 0.014 |
| | (0.503) | (0.524) | (0.498) | (0.515) |
| Lag_mutual_funds | −0.031 | −0.030 | −0.031 | −0.031 |
| | (−1.113) | (−1.092) | (−1.122) | (−1.130) |
| Zero_return_days | 0.005 | 0.008 | 0.005 | 0.006 |
| | (0.217) | (0.306) | (0.184) | (0.227) |
| Effective_spread | 0.020 | 0.021 | 0.020 | 0.020 |
| | (0.732) | (0.775) | (0.715) | (0.737) |
| Percent_bonds | 0.028 | 0.027 | 0.029 | 0.032 |
| | (1.141) | (1.067) | (1.182) | (1.283) |
| Volatility | 0.006 | 0.005 | 0.007 | 0.009 |
| | (0.237) | (0.201) | (0.274) | (0.354) |
| Fair_value | −0.003 | −0.005 | −0.001 | −0.001 |
| | (−0.048) | (−0.097) | (−0.025) | (−0.020) |
| financial_sector | 0.026 | 0.026 | 0.027 | 0.026 |
| | (0.472) | (0.460) | (0.475) | (0.471) |
| Domestic_contagion | −0.025 | −0.032 | −0.020 | −0.012 |
| | (−0.950) | (−1.065) | (−0.693) | (−0.411) |
| Global_crisis_08 | 0.001 | 0.003 | −0.002 | −0.007 |
| | (0.107) | (0.481) | (−0.261) | (−0.760) |
| Global_crisis_11 | 0.005 | −0.002 | −0.011 | 0.036 |
| | (0.610) | (−0.217) | (−0.712) | (0.899) |
| Spillover_market | | −0.024 | 0.017 | 0.040 |
| | | (−0.489) | (0.369) | (1.016) |
| Adj. $R^2$ | 0.835 | 0.835 | 0.835 | 0.836 |

In all the tables, *** Significant at the 1% level.

With regard to the effect of creditor monitoring, hypothesis H5 states that the holdings of local pension funds, mutual funds, and retail investors are sensitive to the level of public debt issued by Chilean firms. The creditor monitoring proxy Percent_bonds is positive and significant in Table 3 above for pension funds. The mutual fund models in Table 4 contains estimates on Percent_bonds that are positive and significant across all models (at $\beta = 0.094$ to 0.103, $p < 0.05$). However, the results for retail investor holdings in Table 5 reveal that the estimate on Percent_bonds is negative and significant (at $\beta = -0.083$, $p < 0.01$ across all models). Thus, we conclude that the results provide support for hypothesis H5. It seems that retail investors held fewer shares in companies with more debt, perhaps

due to imperfect information that led them to infer a higher probability of default. In contrast, mutual funds and pension funds (marginally significant) held more shares in firms with more debt in the capital structure, a result perhaps related to the value they place on the monitoring of management by creditors, an important external corporate governance mechanism.

## 6. Discussion

In Table 4, for mutual fund holdings, the Financial_sector dummy enters all the models as negative and significant (i.e., $\beta = -0.235$, $p < 0.01$), but positive and significant across all models for retail investors in Table 5 (i.e., $\beta = 0.10$, $p < 0.10$). A study of the banking sector firms' market reaction to the fair value accounting (FVA) and impairment rules during the 2008 crisis by Bowen and Khan (2014, p. 233) found that "investors acted as if the potential negative effects of then-existing FVA and impairment rules outweighed any benefits associated with having more timely and transparent mark-to-market data for decision making". Thus, it may be that risk-averse mutual fund managers avoided shares of financial institutions during these two crisis periods. This could be related to contagion if mutual fund managers were biased away from investing in financial institutions, given the uncertainty surrounding their exposure to US mortgage-backed securities and EU sovereign debt defaults. Thus, while creditor monitoring may have motivated institutional investors to increase equity holdings during a period highlighted by two severe GFC's, uncertainty regarding the precise nature of assets on the balance sheets of financial institutions and a failure to transparently recognize impairment losses on their investments may have had a negative and offsetting effect. Therefore, weaker regulation of firms within the financial sector may have hastened the outflow of funds through a reduction in mutual fund holdings across the study period.

Table 7 below compares our generalized spillover and contagion results to the Chilean results in studies by Bekaert et al. (2014) and Dungey and Gajurel (2015). Table 7 shows that our results for retail investors provide a possible source for the Bekaert et al. (2014) and Dungey and Gajurel (2015) findings for Chilean contagion during the 2008 crisis. Moreover, there is no evidence that the 2011 GFC had any contagion effects on Chile's equity market, which we attribute to improved regulations, creditor monitoring, and fair value measurement following IFRS adoption.

We provided insights into potential herding behavior among the four investor groups, by considering the lagged values of ownership holdings. Table 3 indicates that across all models, the current period holdings of pension funds are a positive and a marginally significant function of the prior period's mutual fund holdings (e.g., $\beta = 0.051$, $p < 0.10$ on Lag_Mutual_funds). This result supports the possibility that local pension funds or AFP managers base at least a portion of their trading decisions on the prior period's mutual fund trades, which we note in Table 7 as pension funds' herding behavior. Brown et al. (2013) attribute institutional herding to reputational effects. However, in the Chilean context, it is unlikely that the AFP managers with perhaps superior information on the Chilean economy were motivated by reputational effects to follow mutual funds. Thus, our results are probably not consistent with Brown et al.'s (2013) study. For mutual funds, the Chilean equity market may have provided a relatively less risky and fundamentally sound investment opportunity compared to global equity markets in the US and EU. Kabir (2018) studied herding in the context of the US financial sector and found evidence for "spurious" herding, which he defined as unintentional herding driven by fundamental factors. Given the Chilean institutional context, spurious herding, as suggested by Kabir (2018), rather than a reputation-based explanation, probably drives our result for AFP managers' herding behavior.

In Table 5, a relationship exists between retail investors and lagged insiders, but it is a negative function of Lag_insider holdings ($\beta = -0.136$, $p < 0.01$ across all models). A possible explanation is that when insiders trade shares, retail investors are on the other side of the trade, hence the negative sign on Lag_Insiders. However, the result is not contemporaneous and warrants further study in future emerging market contagion research.

**Table 7.** Comparing this study's result coefficients for spillover and contagion with other studies (pos = positive, neg = negative).

| | Summary of Ownership and Contagion Results of This Study | | | | Returns and Contagion Studies | |
|---|---|---|---|---|---|---|
| **Factor** | **Pension Funds** | **Mutual Funds** | **Retail Investor** | **Insiders** | **Bekaert et al. (2014)** | **Dungey and Gajurel (2015)** |
| d_factor | pos ** | 0 | 0 | 0 | | |
| g_ret | neg ** | 0 | 0 | 0 | | pos |
| e_ret | neg ** | 0 | 0 | 0 | | |
| u_ret | 0 | 0 | 0 | 0 | | neg |
| d_gfc_2008 | neg ** | neg ** | pos *** | 0 | pos | |
| d_gfc_2011 | 0 | 0 | 0 | 0 | | |
| g_gfc_2008 | neg *** | 0 | pos * | 0 | pos | pos |
| g_gfc_2011 | 0 | 0 | 0 | 0 | | |
| e_gfc_2008 | neg *** | neg ** | pos * | 0 | pos | |
| e_gfc_2011 | 0 | 0 | 0 | 0 | | |
| u_gfc_2008 | neg ** | 0 | pos * | 0 | pos | neg |
| u_gfc_2011 | 0 | 0 | 0 | 0 | | |
| herding | pos *, with Mutual funds | no | neg ***, with Insiders | no | yes | |

In all the tables, *** Significant at the 1% level, ** significant at the 5% level, and * significant at 10%.

## 7. Conclusions

In this study, the methodology of Bekaert et al. (2014) and Dungey and Gajurel (2015) was modified to facilitate an analysis of contagion, creditor monitoring, and herding in terms of equity holdings instead of equity returns during a period that includes the 2008 US mortgage crisis and the 2011 EU debt crisis. One conclusion was that publicly traded shares of Chilean firms became more concentrated in the hands of mutual funds and pension funds during the 2008 US mortgage crisis, and less concentrated in the hands of retail investors. Thus, retail investors served as a mechanism to transmit contagion to Chile's stock market during the 2008 global financial crisis, while changes in the equity holdings of institutional investors tended to mitigate the transmission of the contagion. However, the same result does not apply to the 2011 EU debt crisis. None of the four investor groups' equity holdings served to mitigate or magnify contagion effects during the 2011 crisis, and we suggested possible reasons for it. A second finding was of potential herding behavior in the Chilean equity market. The results indicate that variation in lagged mutual fund holdings explain a statistically significant portion of the variation in current period local pension fund or AFPs holdings during this period.

Another result pertains to the relationship between creditor monitoring and the equity holdings of institutional investors. Results show that high levels of publicly issued debt in the capital structure of Chilean firms are associated with higher pension fund and mutual fund equity holdings. In contrast, the equity holdings of retail investors are lower in companies with higher levels of publicly issued debt. It appears that during a period defined by two substantial financial crises, institutional investors favored firms with more public debt in their capital structure, while retail investors may have associated a higher probability of default with more debt. The result suggests that in emerging markets during crisis periods, institutional investors prefer to invest in companies with greater potential for creditor monitoring of management, while retail investors may fear a greater risk of default. Thus, it does not automatically follow that firms with greater debt are more likely to have greater outflows.

Some local institutional investors have a relatively steady flow of funds regardless of market conditions. This is particularly true of Chilean pension funds, given that payroll deduction is the source of their investable funds. It is plausible that greater opaqueness prior to IFRS adoption in 2009 led institutional investors to continue to invest in Chilean equities at historical rates during the

2008 crisis, but not during the post IFRS adoption period containing the 2011 EU crisis. In other words, it is possible that an increase in transparency associated with fair value reporting allowed fund managers to differentiate companies by risk and adjust their investment strategies accordingly, particularly in the financial sector. If true, then IFRS adoption during the study period may explain the finding that institutional investors mitigated contagion during the 2008 crisis, but not during the 2011 crisis. A major limitation of the study is that it covers only the Chilean stock market that has less than 70 actively traded firms. Nevertheless, the conjectures that we raised warrant further analysis, and we encourage future research to pursue this line of investigation to identify the sources of contagion transmission into other emerging capital markets by extending Bekaert et al.'s (2014) model and studying the benefits of having an active bond market.

**Author Contributions:** All authors contributed equally. Conceptualization, S.M.; Data curation, T.G.; Formal analysis, S.M.; Investigation, B.S.-P.; Methodology, T.G.; Project administration, B.S.-P. All authors have read and agreed to the published version of the manuscript.

**Funding:** This research received no external funding.

**Acknowledgments:** The authors would like to thank the Chilean Stock Exchange for providing us the data. We would also like to thank the regulators and research staff of the Chilean Santiago Stock Exchange and Superintendecia Valores y Seguros that is now referred to as the Commission for the Financial Markets for their valuable insight during the presentation at their offices in Santiago, Chile.

**Conflicts of Interest:** The authors declare no conflict of interest.

## Appendix A

**Table A1.** Study Variables and Definitions.

| Variables | Definitions |
|---|---|
| insiders | The percentage of outstanding shares held by the top 10 shareholders. |
| pension_funds | The investment of AFP's measured as the percentage of shares held by local pension funds. |
| mutual_funds | The percentage of outstanding shares held by all other institutional investors. |
| retail | The percentage of outstanding shares not held by insiders and institutional investors. |
| pct_bonds | The percentage value of publicly issued bonds to total equity. |
| spillover_market | The annual return for each of three foreign equity markets: |
| | EU Spillover—the annual return for the "MSCI European stock market Index", |
| | Global Spillover—the annual return for the "MSCI World Index", and |
| | US Spillover—the annual return for the "MSCI US stock market Index". |
| Global_crisis_08 | Dummy variable that equals spillover_market in 2008, 0 otherwise. |
| Global_crisis_11 | Dummy variable that equals spillover_market in 2011, 0 otherwise. |
| Domestic_contagion | "domestic factor"—the orthogonalized annual return of the Chilean equity market against EU Spillover, Global Spillover, and US Spillover. |
| Control Variables | |
| volatility | The annual average standard deviation of the daily stock return for each company. |
| zero_return_days | The number of each company's trading days per year with a 0% return. |
| fair_value | 1 if the firm uses fair value reporting, 0 otherwise. |
| fin_dum | 1 if the firm is in the financial sector, 0 otherwise. |
| es | Effective spread—trading cost defined as: Trade execution price-midpoint of quoted spread divided quoted spread. |

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
