# Peer review of "Stock Market Contagion during the Global Financial Crises: Evidence from the Chilean Stock Market"

_ijfs, doi:10.3390/ijfs8020026_

Round 1

Reviewer 1 Report

Excellent article. Some minor grammatical checks required. eg; "The herd behavior' - 'The' is not required.

Introduction and literature review is very long. Could be significantly shortened. As could methodology

The tested equations are never stated. These should be listed and linked to hypotheses.

It is not normal to use a negative hypotheses. This needs justifying. 

It is normal style to briefly state main points of a table under the heading of that table.

Author Response

Authors Response to Comments and Suggestions from Reviewer 1 

Excellent article. Some minor grammatical checks required. eg; "The herd behavior' - 'The' is not required.

Below, the reviewer comments are stated first, followed by the authors’ replies.

Authors Reply: Thanks you for a positive review.  We have revised “The heard behavior” statement and made many other corrections to the grammatical errors, a number of which were pointed out by two other reviewers.

Introduction and literature review is very long. Could be significantly shortened. As could methodology.

Authors Reply: We have revised the introduction section, and made an effort to shorten the literature review and methodology sections too.

The tested equations are never stated. These should be listed and linked to hypotheses.

Authors Reply: In keeping with the goal of having a shorter methodology section, we decided not to include additional equations. Instead, we added the following paragraph below that explains how Equation (1) was modified to test the hypotheses.  We believe this revision will help explain the link between the hypotheses and the equations tested without unnecessarily lengthening the methodology section.

“To test the hypotheses, four models are estimated for insiders, mutual funds, pension funds and retail investors that complement the studies of Dungey and Gujral, (2015) and Bekaert et al. (2014). However, as shown above, Equation (1) replaces their dependent variable based on equity returns with the equity holdings of insiders, mutual funds, pension funds and retail investors. An analysis based on equity holdings facilitates an examination of the possibility that institutional managers, retail investors and insiders provide liquidity when market makers pull back from the provision of providing liquidity during GFC’s.”

It is not normal to use a negative hypotheses. This needs justifying.

Authors Reply: Thanks for making this point.  In order to make our hypotheses clearer we have included the following statement (just before the hypotheses) in section 4.1.

            “Our models separate the equity market response to the two GFC’s into changes in the equity holdings of four distinct shareholder groups: insiders, local pension funds or AFP’s, mutual funds and retail investors. In this context, to provide directional hypotheses, we state our hypotheses in terms of the type of investor whose holdings are likely to be a mechanism for transmitting the contagion during a GFC, and those that are not likely to be mechanisms for transmitting the contagion”. 

It is normal style to briefly state main points of a table under the heading of that table.

Authors Reply: We are reluctant to state the main points of each table under the heading because we couldn’t simplify the main points consistently across the different tables (e.g., Tables 3 – 6).  Further, in the results section, in an earlier paragraph we had explained the tables and the purpose of the models tested consistently across each table.  Hence, we felt such a general explanation of the tables should better focus the reader on the differences in behaviors of the different equity holders than providing inconsistent main points for each table.  

Reviewer 2 Report

I have no major concerns, but a couple of suggestions.

The "introduction" was essentially the first 10 pages. Could this be made more conciuse and get to the heart of the paper earlier?

There are numerous minor errors in the text that should be cleaned up.

For example:

(72) a majority

(78) insiders not insider's

(83) The US

(131) Section five

(146) effects of s GFC ...

(151) GRC's (not sure if using the possessive makes sense here and AFP's used later)

(475) and (484) the respectively is never needed

(542) can delete a (

Author Response

Authors Response to Comments and Suggestions from Reviewer 2

Comments and Suggestions for Authors:

I have no major concerns, but a couple of suggestions.

  • Below, the reviewer comments are stated first, followed by the authors’ replies.

The "introduction" was essentially the first 10 pages. Could this be made more concise and get to the heart of the paper earlier?

Authors Reply: In the revised manuscript, we get to the heart of the paper by page 3, which is the introduction section.  We have also done our best to shorten the sections 2 and 3 that are the Chilean Institutional Context and the Literature Review sections, respectively.

There are numerous minor errors in the text that should be cleaned up.

For example:

(72) a majority

(78) insiders not insider's

(83) The US

(131) Section five

(146) effects of s GFC ...

(151) GRC's (not sure if using the possessive makes sense here and AFP's used later)

(475) and (484) the respectively is never needed

(542) can delete a (

 Authors Reply: Thanks for pointing out these errors, and sorry we did not catch them earlier.  We have fixed the above errors and carefully revised and proof edited the paper (with the help of a colleague) making sure no other errors exists now.

Reviewer 3 Report

The paper is well written and in genereal well executed. I have only some minor comments:

Page 4, Row 148: It would be informative to extend the time period of the graphs as it is very difficult to understand if the differences are due to GFC or not otherwise.
1.P 11, R 336: "As a first step, each of the four equity index returns is orthogonalized by regressing it against the other three returns." The problem here is that this implies that the residual is only orthogonal to the other indices and not the residuals of those regressions. Further, this operationalisation does not correspond to the definition in Row 331 "Spillover refers to the average variation in equity holdings associated with variation in external equity markets".
2.P 12, note 3: The Durbin-Watson test is an extremely bad test when testing for autocorrelation. It has lower power than many other tests and further, the critical values depends on the explanatory variables in the regression.
3.There is a technical problem as the dependent variable is a percentage. If that variable is relative far from the boundaries, there is no problem. The problem occurs if the dependent variable is close to the boundaries 0 and 100% as then the error becomes skewed and with expectation not zero. This has serious consequences on the parameter estimates if one is unlucky.
4.Row 493: If there is high multicollinearity then this implies that there is difficulties separating the effect from the individual variables. By running separate regressions you do not circumstance this fact, i.e. you cannot interpret the coefficients in the ceteris paribus interpretation as you usually do in a regression.

Author Response

Authors Response to Comments and Suggestions from Reviewer 3

Comments and Suggestions for Authors:

The paper is well written and in general well executed.

I have only some minor comments:

  • Below, the reviewer comments are stated first, followed by the authors’ replies.

Page 4, Row 148: It would be informative to extend the time period of the graphs as it is very difficult to understand if the differences are due to GFC or not otherwise.

Authors Reply: We attempted to obtain additional data, but could not.  A challenge is that the co-author who supplied the data has retired, and so far we have been unable to obtain all the data necessary to extend the study, particularly given the short-time period within which we had to respond to the reviewer comments.

1.P 11, R 336: "As a first step, each of the four equity index returns is orthogonalized by regressing it against the other three returns." The problem here is that this implies that the residual is only orthogonal to the other indices and not the residuals of those regressions. Further, this operationalisation does not correspond to the definition in Row 331 "Spillover refers to the average variation in equity holdings associated with variation in external equity markets".

Authors Reply: The objective is to use the residuals as factors that isolate the portion of variation in in the domestic factor that is not explained by the other three indices.  By design, this type of regression accounts for the covariation between the “y” indice and the other “x” indices through the parameter estimates on the x indices.  The residuals of the regression represent variation in the y index that is not explained by the x indices.  For example, if the domestic index is the y variable, the residuals of the regression on the “external” x indices (Global, European and US) contain the portion of the variation in y(domestic) that is not explained by the x’s (Global, European and US indices). Continuing with the example, the second step is to use the residuals from the domestic index regression as an explanatory variable in the ownership regressions.  If the coefficient estimate on the residual is significant, then the implication is that the impact of the domestic factor on ownership has been isolated from the impact of the Global, European and US factors. Here is an explanation from a previous study, Baekart et.al.(2014)-                                                                                                                                        

In order to obtain an intuitive interpretation of the estimates of the factor loadings, we orthogonalise the three factors. The global factor is orthogonalised by regressing global financial sector returns on US returns over the full sample period (including the crisis period) and then using the residuals of this regression as the global factor. Similarly, following Bekaert, Hodrick and Zhang (2009), we extract a domestic return component which is orthogonal to those of both the US factor and the global factor by regressing each domestic market return on US returns and global financial sector returns, and then using the residual of this regression as the domestic factor.   

And in the revised manuscript we have made an effort to explain this process much more clearly too.

2.P 12, note 3: The Durbin-Watson test is an extremely bad test when testing for autocorrelation. It has lower power than many other tests and further, the critical values depend on the explanatory variables in the regression.

Authors Reply: Thank you for noting this point. This was on oversight on our part.  Fortunately, after changing to the Durbin-h test to account for lagged dependent variables in our regressions, we find the same significance for autocorrelation, and we were able to continue to use the AR1 models to generate results in the paper. We have revised footnote 3 accordingly too.

  1. There is a technical problem as the dependent variable is a percentage. If that variable is relative far from the boundaries, there is no problem. The problem occurs if the dependent variable is close to the boundaries 0 and 100% as then the error becomes skewed and with expectation not zero. This has serious consequences on the parameter estimates if one is unlucky.

Authors Reply: This potential shortcoming is acknowledged. We use percentages to remain consistent with the existing ownership literature studies.  Additionally, in our specific case, the percentages are well-distributed across the 0-1 range for the ownership of insiders and retail, but less so for the ownership of mutual funds and pension funds.  However, the inspection of the regression residuals indicates they are normally distributed with somewhat higher peaks for mutual funds and pension funds relative to the distribution of residuals for insiders and retail that we don’t believe affects the results.

  1. Row 493: If there is high multicollinearity then this implies that there is difficulties separating the effect from the individual variables. By running separate regressions you do not circumstance this fact, i.e. you cannot interpret the coefficients in the ceteris paribus interpretation as you usually do in a regression.

Authors Reply: Thank you for this insight.  We have noted note it in our revised manuscript, and edited the interpretations in the results section so it is clear that substantial collinearity between the external spillover markets implies that the underlying signal from each specific external market contains the influence of the other two external markets.